# Enhancing Genomic Prediction Accuracy of Reproduction Traits in Rongchang Pigs Through Machine Learning

**DOI:** 10.3390/ani15040525

**Published:** 2025-02-12

**Authors:** Junge Wang, Jie Chai, Li Chen, Tinghuan Zhang, Xi Long, Shuqi Diao, Dong Chen, Zongyi Guo, Guoqing Tang, Pingxian Wu

**Affiliations:** 1Farm Animal Genetic Resources Exploration and Innovation Key Laboratory of Sichuan Province, Sichuan Agricultural University, Chengdu 611130, China; wjgdms@163.com (J.W.); 2020302162@stu.sicau.edu.cn (D.C.); 2Chongqing Academy of Animal Sciences, Chongqing 402460, China; 13883643003@163.com (J.C.); 18728165696@163.com (L.C.); 15683401600@163.com (T.Z.); 19922803335@163.com (X.L.); saradiao@126.com (S.D.); guozongyi@163.com (Z.G.); 3National Center of Technology Innovation for Pigs, Chongqing 402460, China

**Keywords:** genomic prediction, machine learning, Rongchang pigs, prediction accuracy

## Abstract

In this study, we investigated the efficacy of machine learning techniques in enhancing genome-wide genomic prediction in Rongchang pigs. Our study focused on designing datasets with varying SNP densities, predicting three reproduction traits, and comparing the performance of traditional methods with five machine learning methods. Our findings demonstrated that as the SNP density increased, both traditional and machine learning methods initially exhibited enhanced predictive performance, with machine learning methods showing improvements ranging from 0.4% to 4.1% across different traits. Furthermore, our study highlighted the unique advantages of machine learning methods, including superior generalizability, computational efficiency, and enhanced prediction accuracy in independent tests. We also discussed the challenges posed by the “curse of dimensionality” and explored strategies to address these challenges, such as downscaling data and identifying key SNPs. In conclusion, our study presents a comprehensive analysis of genome-wide genomic prediction in Rongchang pigs, emphasizing the role of machine learning techniques in improving prediction accuracy and efficiency. We believe that our research contributes valuable insights to the field of genomic prediction and holds implications for advancing local pig genome breeding practices.

## 1. Introduction

Currently, genome-wide prediction (GP) technology has been widely used in plant and animal breeding [1,2,3]. GP involves constructing statistical models based on the correlation between the genotypes of molecular markers and their associated phenotypes. Subsequently, it facilitates the accurate prediction and selection of breeding populations with unknown phenotypes [1]. In the field of livestock and poultry breeding, GP has been successfully utilized in several species, facilitating breeding efforts [4,5,6,7,8]. Commonly, GP is categorized into Genomic Best Linear Unbiased Prediction (GBLUP) and Bayesian models, wherein the primary distinction lies in the underlying a priori distributions of marker effects [1]. Due to advancements in sequencing technologies, acquiring detailed genomic data from individual subjects has become increasingly accessible. This theoretically enhances the accuracy of GP. However, a prevalent challenge in GP implementation is the “small n large p” issue, where the number of samples (n) in the analyzed population is significantly smaller than the number of markers (p) [9]. This disparity leads to errors in estimating marker effects, thereby impacting the final predicted outcomes. In addition to the large increase in genomic data, GP is frequently applied to populations with limited sample sizes, exacerbating the aforementioned challenges [10]. Various models and algorithms have been proposed to address these issues through dimensionality reduction techniques like LASSO (Least Absolute Shrinkage and Selection Operator) and RKHS (Reproducing Kernel Hilbert Space) [11,12]. Furthermore, in addition to the “small n large p” dilemma, the emergence of multi-omic data has yielded a wealth of a priori information [13,14]. However, traditional linear-based GP models primarily focus on additive effects, often overlooking potential nonlinear relationships between markers and phenotypes [2]. To enhance the efficacy of GP, novel analytical strategies should be further explored to unlock the full potential of genomic data.

With the continuous advancement of computer technology, machine learning (ML) techniques have attracted significant attention in the past decade. One of the notable attributes of ML methods is their capability to handle nonlinear relationships within high-dimensional data by utilizing specific model constructs, thereby exhibiting high computational efficiency [15]. In contrast to traditional GP, ML does not rely on prior assumptions regarding marker effects. Instead, it assigns effect values during model training, resulting in discrepancies between the outcomes of ML and traditional GP. Srivastava et al. [16] demonstrated that extreme gradient boosting exhibited superior predictive performance for carcass weight and marbling score in Hanwoo cattle. Similarly, Alves et al. [17] identified support vector regression as the optimal method for predicting reproduction traits in Nellore cattle. For different traits, the average prediction accuracy ranged from 0.27 to 0.67. Moreover, Wang et al. [8] compared various ML techniques with BLUP and Bayesian methods, illustrating that ML methods are more effective in predicting relationships among reproduction traits in pigs. ML techniques have been widely applied in diverse species, including sheep and aquatic animals [6,18,19], as well as in plants and microorganisms [7,20,21,22,23]. Despite the extensive research on ML in genomic prediction, it is noteworthy that no single method is universally suitable for all species or traits. Different methods exhibit varying performances under distinct circumstances, with instances where ML may not outperform traditional methods [2,24,25,26].

Data generation in biology and biotechnology has greatly increased in recent years due to the very rapid development of high-performance technologies [27]. These data are obtained from studying biological molecules, such as metabolites, proteins, RNA, and DNA, to understand the role of these molecules in determining the structure, function, and dynamics of living systems [27]. Functional genomics is a field of research that aims to characterize the function and interaction of all the major components (DNA, RNA, proteins, and metabolites, along with their modifications) that contribute to the set of observable characteristics of a cell or individual (i.e., phenotype). Furthermore, in a breeding program, genetic improvement can be maximized through the accurate identification of superior animals that are selected as parents of the next generation, thereby achieving breeding goals [28]. Artificial neural networks have been proposed to alleviate this limitation of traditional regression methods and can be used to handle nonlinear and complex data, even when the data are imprecise and noisy). Omics data can be too large and complex to handle through visual analysis or statistical correlations. This has encouraged the use of machine intelligence or artificial intelligence [28].

The use of ML offers numerous advantages when analyzing high-dimensional data. However, it is important to acknowledge the challenges posed by the “curse of dimensionality” especially when dealing with small populations or large sequencing datasets [10,29]. To address this issue, various strategies can be employed to downscale the data, such as utilizing LD-based haplotypes [30,31], identifying key single nucleotide polymorphisms (SNPs) [18,32], or developing new analytical models [33,34]. These approaches all aim to strengthen the role of key information in model fitting [35].

This study focuses on exploring solutions to the aforementioned challenges by examining the Rongchang pig, a local Chinese pig breed. The objectives include (1) comparing the impact of different data dimensions on GP; (2) assessing the effect of the priori information on ML; and (3) comparing ML techniques with traditional GP methods to determine the most suitable approach. Our findings may provide a method for local pig genome breeding that can perform rapid genomic prediction at high SNP densities. Using the ML for genomic prediction can help address the issues of small population size and SNP diversity in local pigs.

## 2. Materials and Methods

### 2.1. Animals and Phenotype

We studied the Rongchang pig population located at the Rongchang Pig National Core Breeding Farm (Rongchang, Chongqing, China). This population comprised 550 individuals, and data on reproduction traits were collected between 2020 and 2022, including litter weight (LW), total number of piglets born (TNB), and number of piglets born alive (NBA).

To ensure the accuracy of the results and minimize the impact of external factors, the recorded phenotypes underwent correction using a linear model. The construction of the linear model was achieved using R 4.3.1. The correction model is represented as follows:(1)y=Xb+e.
where y denotes the original phenotypic data, X represents the matrix incorporating the effects into the model, b stands for the fixed-effects matrix containing factors to be excluded such as year, month, and total litter size of sows, and e signifies the residuals. After calculating e, any aberrant phenotypes were eliminated, and the data from the remaining 515 Rongchang pigs, denoted as $y_{c}$, were standardized for subsequent analysis.

### 2.2. Genomic DNA Extraction

Ear tissue samples (n = 550) were obtained and preserved in 75% alcohol. All individuals are female. Genomic DNA was extracted from these ear tissue samples utilizing the Tissues Genomic DNA kit (Omega Bio-Tek, Norcross, GA, USA) as per the manufacturer’s protocol. Subsequently, the quality and quantity of the genomic DNA were assessed using a Nanodrop-2000 spectrophotometer (Thermo Fisher Scientific, Wilmington, NC, USA). DNA samples meeting the criteria of a light absorption ratio (A260/280) between 1.8 and 2.0, concentration ≥ 50 ng/µL, and a total volume ≤ 50 µL were considered suitable for further analysis.

### 2.3. Genotyping Using Zhongxin-1 Porcine Breeding Array PLUS

A total of 550 samples were genotyped using the Zhongxin-1 Porcine Breeding Array PLUS (Compass Biotechnology, Beijing, China), which comprised 57,466 SNPs. Initially, SNPs with no positional information and those located on sex chromosomes were eliminated from the genotype data, resulting in 9435 SNPs being removed. Subsequently, quality control of the genotype data was conducted using the PLINK (v1.9) software. SNPs with call rates below 0.95, minor allele frequencies (MAF) below 0.01, and Hardy–Weinberg equilibrium test *p*-values below 10^−6^ were excluded from the dataset. Following quality control measures, a total of 16,275 SNPs were removed, leaving 31,756 SNPs as the final target genotype dataset.

### 2.4. Whole-Genome Sequencing and SNP Calling

Whole-genome sequencing reference data were obtained for 120 Rongchang pigs, utilizing 150 bp paired-end reads on the BGI platform in Wuhan, China, employing the PE150 strategy. The sequencing was conducted by BGI Co., Ltd., in Wuhan, China. After sequencing, the quality of raw reads was assessed using a minimum Phred score of 20 to filter out adapter-polluted reads and reads containing an excessive number of ‘N’ characters (where ‘N’ constitutes more than 10% of a single read) to generate clean reads, as evaluated by FastQC (https://www.bioinformatics.babraham.ac.uk/projects/fastqc/, accessed on 14 October 2024). Then, the clean reads were aligned to the pig reference genome (Sscrofa11.1) utilizing BWA software (version 0.7.15) with the parameters ‘mem -t 10 -k 32 -M’. The resulting SAM files were transformed into BAM files using SAMtools (version 1.19). Potential PCR duplicates were eliminated using the MarkDuplicates utility within Picard release 1.119 (https://sourceforge.net/projects/picard/files/picardtools/1.119/, accessed on 14 October 2024). Subsequently, the BAM files were utilized to call SNPs with the GATK (version 3.5) software employing multi-sample approaches. The raw SNPs generated by GATK underwent further filtering based on the following criteria: QualByDepth < 2.0, FisherStrand < 60.0, RMSMappingQuality < 40.0, MappingQualityRankSumTest < −12.5, and ReadPosRankSumTest < −8.0. Following this filtering step, a total of 21,104,245 SNPs remained. For subsequent analyses, SNPs with MAF ≥ 0.05, missing rates < 10^−6^, read depths ≥ 6, and SNPs located on autosomes were considered as the reference genotype data.

### 2.5. Imputation from 50 K Chip to Whole-Genome Sequencing Data

Genotype imputation between the target and reference genotype data was conducted using Beagle (version 5.4). The reference sequence consisted of 30× resequencing data from 90 Rongchang pigs. To assess imputation accuracy, 30 Rongchang pig individuals were randomly selected from the imputation population, and their corresponding 30× resequencing data were obtained. The genotypes of each locus were compared, and any locus with incorrect phasing was considered correctly populated through this comparison process. Subsequently, the imputation accuracy of the locus was calculated as the ratio of the number of correctly populated individuals to the total number of individuals at that locus. Following imputation, SNPs with an imputation accuracy of 1 were retained. Additionally, SNPs with MAF < 0.01 were excluded from the imputed data. Consequently, a total of 8,823,367 SNPs were retained for further analysis.

### 2.6. Association Analysis

Genome-wide association studies (GWASs) were conducted on 515 Rongchang pigs using PLINK (v1.9) software. The analyses used Principal Component Analysis (PCA) results as covariates. Specifically, 30 PCA analyses were performed for each trait, and the *p* value of each PCA was calculated using EIGENSTRAT. PCAs with a *p* value < 0.01 were used as covariates in the analyses.

### 2.7. Priori Information Processing

Loci associated with the target traits were designated as priori information 1, as per the Pig Quantitative Trait Locus (QTL) database (https://www.animalgenome.org/cgi-bin/QTLdb/SS/index, accessed on 25 November 2024). Loci significantly associated with the target traits were identified as priori information 2, based on the GWAS results. The phenotypic variance explained by a given SNP at each locus was determined using GEMMA (v0.98.5) software. Weights were assigned based on the percentage of phenotypic variance derived from the priori information. A genomic relationship matrix (GRM) was constructed using GMAT (v1.01) software with priori information.

### 2.8. Methods Used for Genomic Prediction

#### 2.8.1. Traditional Methods

Genomic prediction was performed using various methods including Genomic Best Linear Unbiased Prediction (GBLUP), Bayes A, Bayes B, Bayes C, Bayes LASSO (BL), and Bayes Ridge Regression (BRR). The predictions by these methods were implemented using the BGLR-R package (v1.1.3) [36].

#### 2.8.2. GBLUP

The GBLUP model is presented as y=1μ+Zg+e, where y denotes the vector of corrected phenotypes of genotyped individuals. The symbol μ symbolizes the overall mean, while **1** is a vector of 1s. Additionally, g represents the vector of genomic breeding values, e stands for the vector of random errors, and Z is an incidence matrix allocating records to g. The random effects were distributed as follows: $g\sim N(0,G\sigma_{g}^{2})$ and $e\sim N(0,I\sigma_{e}^{2})$, where G represents GRM, $\sigma_{g}^{2}$ is the additive genetic variance, and $\sigma_{e}^{2}$ is the residual variance. Phenotype prediction using the GBLUP method was performed using the BGLR-R package [37].

#### 2.8.3. Bayes Methods

Bayesian analysis involves combining the a priori information of unknown parameters with sample information. The posterior information is then derived using the Bayesian formula, and this posterior information is used to infer the unknown parameters. Posterior statistics for further inference are typically obtained through Markov Chain Monte Carlo after a specified number of iterations under a specific prior condition. Different Bayesian methods vary in their prior conditions. Phenotype prediction was conducted using various Bayes methods with the BGLR-R package [38]. The a priori conditions for these methods are detailed in Table 1 [36].

#### 2.8.4. ML Methods

Five different ML methods, namely kernel ridge regression (KRR), random forest (RF), Gradient Boosting Decision Tree (GBDT), Light Gradient Boosting Machine (LightGBM), and Adaboost, were also used for genomic prediction. The ML models were developed using the Scikit-learn library in Python, and hyperparameters were tuned using Bayesian optimization with 300 iterations.

#### 2.8.5. KRR

KRR is a nonlinear regression technique that effectively captures the underlying nonlinear patterns within the data [33]. KRR operates by applying a nonlinear kernel function to map the data into a higher dimensional space, followed by constructing a ridge regression model to ensure linear separability within this kernel space. The prediction model for KRR can be expressed as:(2)$$y\left(x_{i}\right)=k^{\prime }{\left(K+\lambda I\right)}^{-1}\hat{y}.$$
where λ represents the regularization constant, K denotes the GRM with entries $K_{ij}=K\left(x_{i},x_{j}\right)=\phi \left(x_{i}\right)\cdot \phi {\left(x_{j}\right)}^{T}$, and I is the identity matrix. For N training samples, the resulting kernel matrix is given by(3)$$K=\left[\begin{array}{cccc} K(x_{1},x_{1}) & K(x_{1},x_{2}) & \ldots & K(x_{1},x_{n})\\ K(x_{2},x_{1}) & K(x_{2},x_{2}) & \ldots & K(x_{2},x_{n})\\ \vdots & \vdots & \vdots & \vdots \\ K(x_{n},x_{1}) & K(x_{n},x_{2}) & \ldots & K(x_{n},x_{n}) \end{array}\right].$$

$k^{\prime }=K\left(x_{i},x_{j}\right),j=1,2,3,\ldots ,n$, where nrepresents the number of training samples, and x_i corresponds to the test sample. $k^{\prime }$ can be written as(4)$$k^{\prime }=\left[\begin{array}{c} K(x_{i},x_{1})\\ K(x_{i},x_{2})\\ \vdots \\ K(x_{i},x_{n}) \end{array}\right].$$

#### 2.8.6. RF

RF is an ensemble learning technique that uses the average prediction of multiple decision trees to determine the classification or prediction for a new instance. By aggregating the results of numerous decision trees, RF mitigates the risk of overfitting [39]. The prediction model for RF is as follows:(5)$$y=\frac{1}{M}\sum_{m=1}^{M}t_{m}(\psi_{m}\left(y:X\right)).$$
where y signifies the predicted value of RF, $t_{m}(\psi_{m}\left(y:X\right))$ represents an individual regression tree, and M denotes the total number of decision trees in the forest. The predictions were obtained by flowing the predictor variables through each tree’s structure, with the estimated value at the terminal node being utilized as the final prediction. The predictions from each tree were then averaged to derive the final prediction for unobserved data.

#### 2.8.7. GBDT

GBDT is an iterative algorithm that constructs a collection of weak decision trees, accumulating their results to generate the final predicted output. This method effectively combines decision trees with concepts of integration. The prediction model for GBDT is expressed as [40](6)$$F\left(x,w\right)=\sum_{m=0}^{M}\alpha_{m}h_{m}\left(x,w_{m}\right)=\sum_{m=0}^{M}f_{m}(x,w_{m}).$$
where x denotes the input sample, w represents the model parameters, h signifies the regression tree, and α stands for the weight of each tree.

#### 2.8.8. LightGBM

LightGBM utilizes a similar algorithm as GBDT, with the inclusion of optimized techniques to enhance efficiency. In LightGBM, the Gradient-based One-Side Sampling is used to selectively sample data based on gradient size, reducing the computational load by focusing on samples with significant gradients. Additionally, Exclusive Feature Bundling is employed to merge and bind certain features, further streamlining computation steps [41,42]. While LightGBM shares fundamentals with GBDT, there are differences between the two methods [43].

#### 2.8.9. Adaboost

Initially developed for classification tasks, Adaboost gradually incorporated regression trees for handling regression problems [44]. The prediction model for Adaboost is given by(7)$$y=inf[y\epsilon Y:\sum_{t:f_{t}(x)\leq y}\log\frac{1}{\varepsilon_{t}}\geq \frac{1}{2}\sum_{t}\log\frac{1}{\varepsilon_{t}}].$$
where y represents the predicted value, $f_{t}\left(x\right)$ denotes the prediction of the tth weak learner, $\varepsilon_{t}$ reflects the error rate of $f_{t}(x)$, $\varepsilon_{t}={\bar{L}}_{t}÷\left(1-{\bar{L}}_{t}\right)$, ${\bar{L}}_{t}$ is the mean value of losses, and $L_{t}(i)$ is the error between the actual observed value and the predicted value of the ith predicted individual. $D_{t}(i)$ is the weight distribution of $f_{t}(x)$.

For all five ML methods, the input independent variable was the GRM, and the response variable was the corrected phenotype ($y_{c}$).

### 2.9. Accuracy Assessment

In this study, the accuracy of the predictive models was assessed using correlation coefficients between the predicted phenotypic values and the actual phenotypic values. The correlation coefficient was calculated using the formula(8)$$r=\frac{cov(y_{pred},y_{true})}{\sqrt{var\left(y_{pred}\right)var(y_{true})}}.$$
where r represents the correlation coefficient, $y_{pred}$ is the predicted phenotype value, and $y_{true}$ is the actual phenotype value. A higher correlation coefficient indicates a more accurate model prediction. Out of the total 515 Rongchang pigs, 400 were selected for training the ML models. The correlation coefficients were computed using a fivefold cross-validation approach, and the average was considered as the final accuracy measure.

### 2.10. ML Model Optimization

Various methods can be employed to optimize ML models, including manual tuning, grid search, and Bayesian optimization. Grid search and Bayesian optimization lead to significant improvements in prediction accuracy, with grid search being more computationally intensive than Bayesian optimization [45]. In this study, Bayesian optimization was utilized to tune the hyperparameters of five ML models. The optimization objective for each model was to maximize the correlation coefficient (r), and the optimal hyperparameters were determined after 300 iterations.

### 2.11. Independent Test

The remaining dataset comprising 115 Rongchang pigs was reserved for an independent test, similar to simulating genomic predictions for candidate herds in practical production using the pre-trained models. The results of the two tests were combined to evaluate the differences between the methods.

## 3. Results

### 3.1. Phenotypic Data

Phenotypic data from 515 Rongchang pigs were obtained after screening. Each trait was analyzed using GCTA (v1.94.1) software and the specific results are presented in Table 2.

In the analysis of variance, the sum of squares within was 9435.24, with 1542 degrees of freedom, and the mean square within was 6.12. The F statistic and *p*-value, which is far smaller than the commonly used significance level, confirm the presence of highly significant differences between the groups. The results of the analysis of variance are shown in Table 3.

### 3.2. Evaluation of Sequencing Data and Genotype Imputation

After initial quality filtering, 11.20 TB of clean data were obtained. The mean number of reads per sample was 312,303,572, with a mean effective rate of 99.91% for each sample. The Q20 and Q30 values were 98.02% and 93.82% per individual, respectively.

This study performed genotype imputation from a 50 K SNP chip to whole-genome sequencing data in Rongchang pigs. A total of 31,756 and 21,104,245 SNPs were identified in the target and reference populations, respectively. After genotype imputation, a total of 21,104,245 SNPs with an average accuracy of 0.94 were available for analysis. The imputation accuracy was calculated at various thresholds—0.3, 0.6, and 0.9, resulting in accuracy ranges of 0.930~0.953, 0.934~0.955, and 0.967~0.977, respectively (Figure 1). The high imputation accuracy indicates the adequacy of the data for further analysis. After screening for 100% imputation accuracy and excluding loci with MAF < 0.01, a total of 8,823,367 SNP loci were obtained for subsequent analysis.

### 3.3. GWAS for Three Reproduction Traits

The results of the GWAS analysis for three reproduction traits are presented in (Figure 2). It was observed that significant loci associated with LW were predominantly located on SSC12, while loci linked to TNB and NBA were primarily situated on SSC14 and SSC15. Loci with a significance level of *p* < 1 × 10^−5^ were selected as a priori information, resulting in the retention of 639, 250, and 266 loci for the three respective traits.

A total of 329, 207 and 123 loci were annotated for the three respective traits. For LW, 32 candidate genes were identified within a 1 Mb range upstream and downstream of these loci. Also, 16 and 9 candidate genes were identified within a 1 Mb range for TNB and NBA. The loci used for annotating LW differ from those used for annotating TNB and NBA. There are no studies reporting candidate genes for NBA specifically in pigs, but the proposed mechanisms are reasonable. The results of the annotation are presented in Table 4.

### 3.4. Accuracy of Genomic Prediction in Cross-Validation

When considering the three traits, traditional methods outperformed ML methods in predictions when the number of loci was below 700 k. Specifically, when the number of loci was 600 k, the Bayes Ridge Regression method demonstrated the highest accuracy in genomic prediction, with values of 0.571 ± 0.068, 0.637 ± 0.040, and 0.628 ± 0.052 for LW, TNB, and NBA, respectively. Conversely, when the number of loci was 700 k, ML methods exhibited superior accuracy compared to traditional methods, with RF and GBDT yielding the best results. Notably, ML approaches showed varying degrees of enhancement in different traits relative to traditional methods, with improvements ranging from 0.4% to 4.1%. The results comparing different methods for LW, TNB, and NBA are depicted in Figure 3.

The genomic prediction accuracy initially increased and then decreased with changes in the number of loci. The peaks of genomic prediction accuracy for all three traits were consistently found in the range of 800 k to 900 k, resulting in values of 0.586 ± 0.073, 0.642 ± 0.044, and 0.669 ± 0.029 for LW, TNB, and NBA, respectively. Notably, the GBDT model demonstrated the best overall performance, followed by the RF model. The peaks in genomic prediction accuracy for traditional methods generally appeared earlier, indicating an advantage in cases with a smaller number of loci.

The effects of locus weighting on machine learning varied significantly across different scenarios. Notably, when the number of loci was substantial, the genomic prediction accuracy of the weighted group exhibited more stable fluctuations, suggesting that locus weighting contributes to improved model stability under high loci scenarios. Results comparing weighted and unweighted predictions for different traits are illustrated in Figure 4, Figure 5 and Figure 6.

### 3.5. Accuracy of Genomic Prediction in Independent Test

In the independent test, following a similar protocol to cross-validation but without the cross-validation step, a different prediction population was utilized. For ML, the model with the best hyperparameters identified during cross-validation was employed for prediction. The results from the independent test showed that the predictions for LW and TNB were comparable to those from cross-validation. Notably, the traditional method exhibited superior genomic prediction compared to ML when the number of loci was limited (<700 k). Furthermore, for predicting the NBA, traditional methods consistently outperformed ML. It is worth mentioning that the genomic prediction accuracy of all methods in the independent test fell below that of cross-validation. To assess the consistency between cross-validation and independent test results for various ML methods, correlation coefficients were calculated, ranging from 0.935 to 0.998 (Table 5). This suggests that the ML models trained demonstrated strong generalization abilities.

Figure 7 illustrates the predictive accuracy of different methods during the independent test, focusing on loci with the best results.

### 3.6. Efficiency of ML Computations

The computations were executed on CPU nodes within the Slurm high-performance cluster, operating at 2.60 GHz with 192 GB of RAM. This cluster has 23 CPUs, 32 cores, and 20 threads. Each computation utilizes all threads of a single node. It was observed that ML methods were notably more efficient than traditional approaches. The execution time (in seconds) at the optimal number of loci during cross-validation was utilized to evaluate the computational efficiency of each method. The KRR model had the shortest execution time, followed by the RF model and the LGB model. The Adaboost model and the GBDT model exhibited time consumption similar to that of the GBLUP method (Figure 8). Conversely, the Bayesian method required significantly more time, with a noticeable increase in execution time as the number of loci increased.

## 4. Discussion

Genotype imputation serves to enhance the density of genotype data, with imputation accuracy typically associated with the concordance between the reference and target sequences. Generally, a closer alignment in density and increased overlap between the sequences lead to improved imputation outcomes. The utilization of imputed data in GWAS analyses can introduce certain distortions due to extrapolating differences across arrays, potentially resulting in elevated false positive signals at specific loci [46]. Notably, the Beagle 5.4 software employs linkage disequilibrium for inference, generating a reference metric indicating the squared correlation between the estimated allele dose and the actual allele. In cases of substantial deviation between the quantities of reference and target data, anomalous results may emerge [47]. Hence, this study employed sampling techniques to compare imputation outcomes with actual sequencing data, resulting in an assessment of imputation accuracy. Loci with 100% imputation accuracy were meticulously preserved post quality control for subsequent analyses, thereby mitigating the impact of imputation errors. Noteworthy findings from Quick et al. [48] underscore the cost-effectiveness of sequencing a subset of participants, substantially amplifying imputation quality and GWAS robustness. Hence, we endeavored to retain the maximum number of target data loci while satisfying stringent quality criteria.

In this study, the mean values of LW, TNB, and NBA for the 515 female individuals were 8.11, 10, and 10, respectively. As a local pig breed, the values of TNB and NBA are lower than those of modern hybrid sows [49]. Reproductive traits in pigs typically have low heritability. Zhao et al. [50] indicate that the heritability of common reproductive traits ranges from 0.03 to 0.21. The heritability of LW and NBA in our study were 0.19 and 0.15, respectively, which are close to those reported in other studies [8,50]. The heritability of TNB is 0.21, which is slightly higher than that reported in other studies. We believe this may be associated with covariate and genotype data.

Some candidate genes identified through GWAS annotation have already been proven to be associated with reproductive traits in pigs. For LW, *HS6ST3* is the selection signal related to growth, conformation, health, reproductive performance, and meat quality [51]. *U2*, *SLC7A7*, and *ME3* can directly affect embryo development [52,53,54]. For TNB, *PRKN* can improve oocyte quality and subsequent embryo developmental competence prior to implantation [55]. *PRF1 is* associated with T-cell immunity [56]; this may indirectly affect TNB. Although there have been no reports of candidate genes related to NBA in pigs, *EFHD1* can affect B-cell development [57], which may interfere with the trait in the same way as *PRF1*. *ARHGAP44* has been shown to affect both fertility and immunity in Holstein cattle [58]. The results suggest that obtaining the weighting of SNP loci from GWAS results is feasible. However, it is important to note that controlling the weights of these loci is crucial, as this may lead to issues such as overfitting.

Genomic prediction through ML has been studied in multiple species [6,16,17,18,19]. For decision tree-based ML methods, such as RF and GBDT, their advantage lies in the ability to better interpret the model, as they make predictions through one or a combination of multiple nodes [2,40]. By aggregating the results of numerous decision trees, tree-based models mitigate the risk of overfitting [39]. Wang et al. [8] compared various ML techniques with BLUP and Bayesian when predicting relationships among reproduction traits in pigs; the RF model achieved better results. However, in terms of computational efficiency, tree-based ML methods may be less efficient [59]. Compared to machine learning methods, deep learning, such as neural networks, is considered to have greater potential. But this also means that neural networks require a larger sample size to build the network in order to achieve the desired results [60].

The results of the cross-validation highlight the distinct advantages and disadvantages of ML and traditional methods in various scenarios. GP operates on the fundamental principle of using genomic data to establish a connection between a reference population and a candidate population. Through this linkage, valuable information from the reference population, such as phenotypes or estimated breeding values, is attributed to individuals in the candidate population. It is noteworthy to mention that the addition of more loci does not necessarily equate to higher prediction accuracy. With the increased number of loci, the complexity of the linkage between populations intensifies, potentially introducing more noise, which can adversely affect the prediction accuracy [61]. In our study, the selection of SNP loci to be added in various groups was based on the results of GWAS. As the number of loci continued to increase, the association of the added loci with the target traits weakened, thereby providing limited enhancement to prediction accuracy and, in some cases, even leading to decreased accuracy. Notably, the transition from 100 k to 200 k loci resulted in a significant improvement in prediction accuracies across both ML and traditional methods. This improvement was attributed to the strong association between this specific group of loci and the target traits. The study by Wang et al. [18] also supports the notion that using significant loci enhances prediction accuracy. Moreover, our study observed a consistent trend in the predictive accuracy of ML and traditional methods as the number of loci varied, with minor discrepancies influenced by the specific objects analyzed and methodological characteristics. Beyond the sheer quantity of loci, the distribution of locus weights in the association model emerged as a critical factor affecting prediction accuracy. Notably, the exceptional accuracy observed with 200 k loci was largely due to the significant association of these loci with target traits, which explained a high percentage of phenotypic variation. Consequently, adjusting the weights of loci artificially can significantly impact prediction accuracy. Additionally, we segmented the loci into three categories for our analysis. The first segment comprised common loci remaining after quality control, while the second segment included exceedingly significant loci identified through GWAS analysis. Lastly, the third segment consisted of loci associated with the target trait as per the Pig QTL database. Our analysis indicates that loci from the second and third categories are more closely linked to the target traits and should thus carry more weight in the modeling process. The weighted loci accounted for 0.765, 0.748, and 0.774 of the base dataset’s phenotypic variation explained by SNP, respectively. Adjusting the weights of the weighted loci to 0.7 was performed uniformly, considering the strong association of subsequently added loci with the target traits and the rankings from GWAS analyses. The GMAT (v0.98.5) software facilitated the generation of weighted and unweighted G matrices as input values for ML training. Notably, different models within the base dataset exhibited distinct responses when employing weighted data for prediction, aligning with findings from previous studies [5,24]. Overall, our study underscores the potential of loci weighting to enhance prediction accuracy, although weight adjustments may result in nuanced effects. Notably, as the number of loci increased, the prediction accuracy of weighted data was often lower than that of unweighted data. This discrepancy may arise from the fact that subsequently added loci play a more prominent role in constructing associations. Given the unique genetic makeup of Rongchang pigs as a regional breed, certain loci in their genome sequences may differ from those cataloged in the Pig QTL database. Consequently, our weighting approach might inadvertently downplay the significance of specific loci in prediction. Moreover, beyond the impact on prediction accuracy, we also observed that weighted data exhibited greater stability at higher points, with less variability in predictions. This phenomenon suggests that data weighting could potentially enhance the model’s robustness under varying conditions.

The key difference between traditional and ML methods lies in their approach to modeling data. Traditional methods typically rely on constructing linear models to address problems, whereas ML methods use nonlinear models. In this study, ML methods were trained using nonlinear kernel functions. Notably, while the KRR model incorporates a nonlinear kernel function, it essentially maps nonlinear data to a kernel space to achieve linear separability within that space [33]. Consequently, the KRR model still adheres to the fundamental concept of a linear model in problem-solving, which explains why its predictive accuracy is lower than that of other ML models when analyzing high-dimensional data. Our results revealed that when the number of loci was less than 700 k, traditional methods exhibited higher prediction accuracy, but as the locus count increased, the difference in prediction accuracy between ML and traditional methods gradually diminished. When the number of loci was greater than 700 k, ML methods gradually demonstrated their advantage in dealing with high-dimensional data. As the dataset shifted, the predictive accuracy of the different methods varied, which is consistent with the results of other studies [62,63,64,65]. For similar studies, Wang et al. [8] used ML methods to predict TNB and NBA, and their prediction accuracy (0.293 and 0.258, respectively) was lower than that of this study. On the one hand, this is because they used chip data, and on the other hand, they did not process the data based on GWAS results. He et al. [66] have shown that significant SNPs can influence the results of genomic prediction. It is crucial to acknowledge that no single method can universally apply to all traits. Factors such as breed, environment, and population structure introduce variability in prediction outcomes [67,68]. In this study, we used both cross-validation and independent tests to comprehensively assess the prediction accuracy of the different methods. While cross-validation and independent tests yielded consistent results for LW and TNB, substantial variation was observed in the predictive performance of ML models for NBA. Further analysis attributed this discrepancy to dataset characteristics. The exploration of hyperparameters exposed overfitting issues with the ML model predicting NBA when the locus count exceeded 600 k. It is feasible to adjust hyperparameters in different models to combat overfitting, such as the learning_rate and max_depth. For the KRR model, tuning the gamma is also effective. On the other hand, in populations with a small sample size, significant loci weighting may also lead to overfitting. It may be more reasonable to plan weights based on the importance of loci. Additionally, the modest sample size and low heritability of NBA in this study are factors that likely influenced prediction accuracy. Given that hyperparameter optimization is based on the accuracy of cross-validation predictions, the optimized hyperparameters are expected to yield enhanced performance when applied to subsequent cross-validation runs. Our study utilized a training set of 400 Rongchang pigs and an independent test set of 115 pigs. While the ML methods successfully used the genomic relationship matrix (GRM) for dimensionality reduction, overcoming the challenges posed by the curse of dimensionality remains elusive. Enhancing the generalizability and robustness of ML models represents a primary challenge in GP [29,30,31]. Notably, certain GPs have demonstrated instances where traditional methods outperform ML methods. Hence, tailoring the training of ML models to suit the unique characteristics of the analysis at hand is pivotal for improving predictive outcomes. With regard to computational efficiency, all methods besides the Bayesian approach used the G-matrix for computation, rendering the Bayesian method more time-consuming in comparison. Apart from Bayes, ML methods exhibited a significant speed advantage, with the KRR model emerging as the fastest due to its reliance on a linear model. In contrast, the GBDT model ranked as the slowest, while the GBLUP method displayed speed advantages, particularly in scenarios with a low number of loci. Overall, ML methods offer significant computational efficiency advantages over Bayesian methods, with prediction accuracy comparable to GBLUP. They are worth using, especially when dealing with a large number of loci.

## 5. Conclusions

This study conducted a comparative analysis of various methods for genome-wide prediction using sequencing data from a large-scale local pig population. The results revealed that the number of loci and the weighting of loci played significant roles in influencing the accuracy of genome-wide prediction. Among the ML models examined, the GBDT model exhibited the most superior performance, achieving the highest prediction accuracy of 0.669 ± 0.029, followed by the RF model. This finding suggests that the GBDT model holds promising potential for further refinement and application in local pig genome breeding. At the same time, optimizing the sample size and weighting structure provides space for improving model performance and combating overfitting. Conclusively, this study provides a method for local pig genome breeding that can perform rapid genomic prediction at high SNP densities, which is conducive to further expanding the population size optimization model in future studies.

## Figures and Tables

**Figure 1 animals-15-00525-f001:**
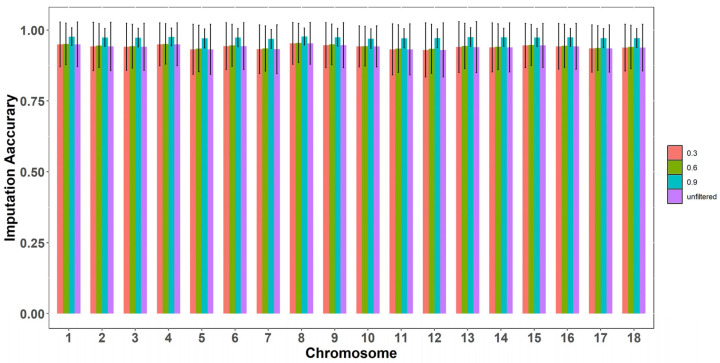
Imputation accuracy for each chromosome in Rongchang pigs. The imputation accuracy at various thresholds—0.3, 0.6, 0.9, and before filtering.

**Figure 2 animals-15-00525-f002:**
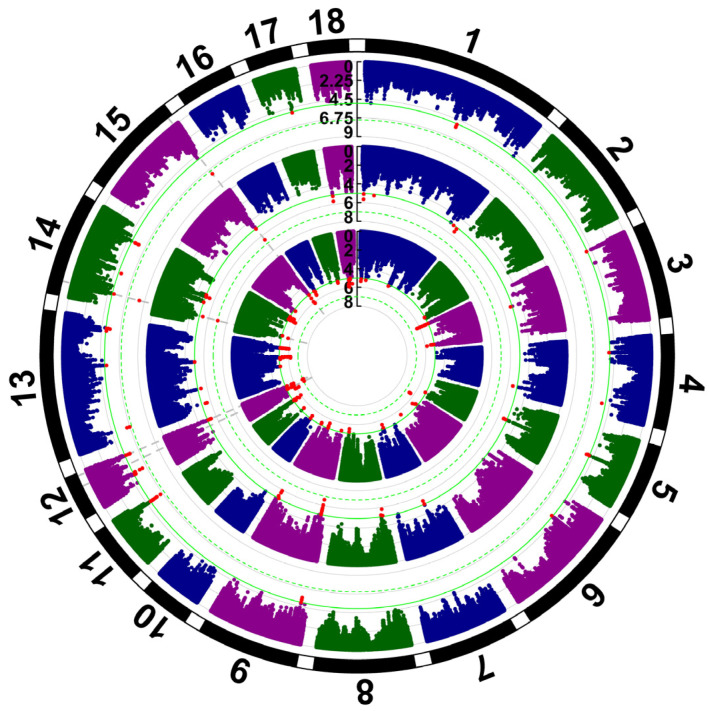
GWAS result for litter weight (LW), total number of piglets born (TNB), and number of piglets born alive (NBA) in Rongchang pigs. LW, TNB, and NBA are shown from inside to outside. Red sites indicate sites used for the weighting of each trait (*p* < 1 × 10^−5^). The numbers on the outer ring indicate chromosomes and the numbers on the central axis indicate −log10(P).

**Figure 3 animals-15-00525-f003:**
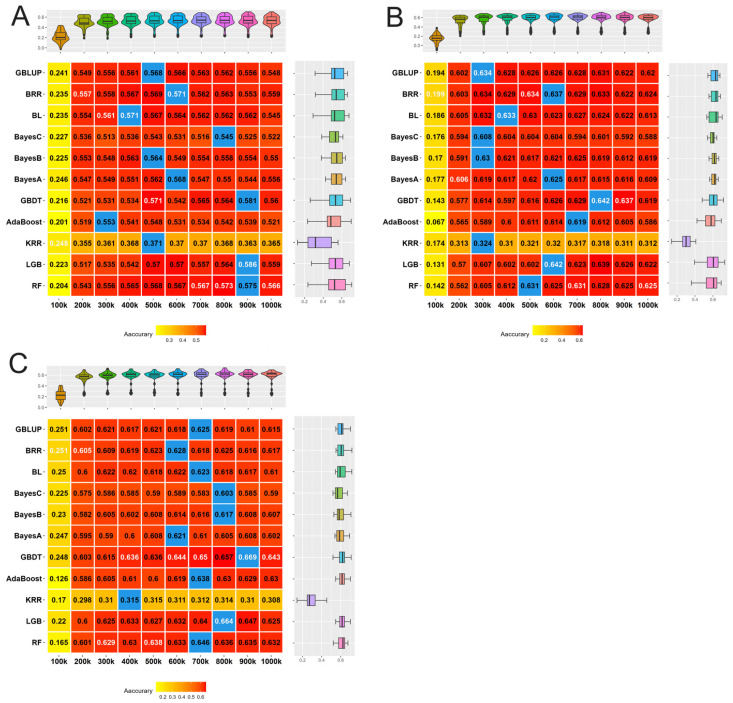
The genomic prediction accuracy of different methods for litter weight (LW), total number of piglets born (TNB), and number of piglets born alive (NBA) in Rongchang pigs. (**A**) The genomic prediction accuracy of LW. (**B**) The genomic prediction accuracy of TNB. (**C**) The genomic prediction accuracy of NBA. The redder the cell, the more accurate the prediction; white text: the highest predictive accuracy of each quantity; blue cell: the highest predictive accuracy of each method. The box plots show the median and distribution of accuracy for each method (right) and the violin plots show the same for each quantity (top).

**Figure 4 animals-15-00525-f004:**
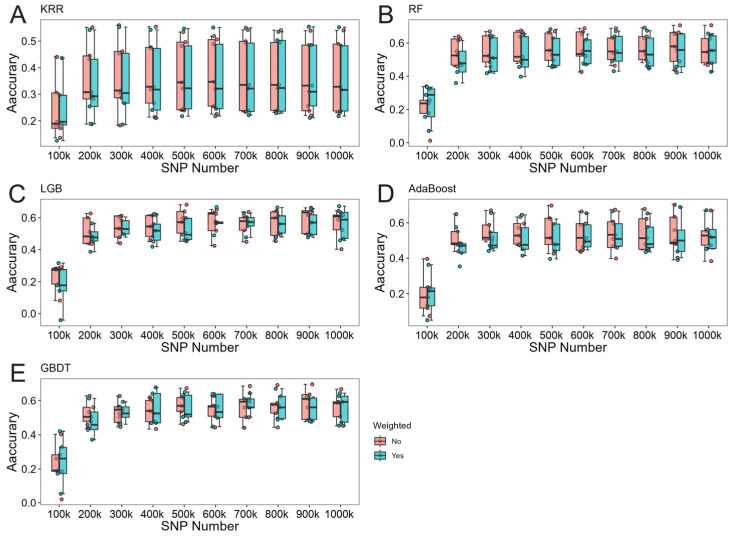
Results comparing weighted and unweighted predictions for litter weight (LW) in Rongchang pigs. (**A**) The predictive results of KRR. (**B**) The predictive results of RF. (**C**) The predictive results of LightGBM. (**D**) The predictive results of Adaboost. (**E**) The predictive results of GBDT.

**Figure 5 animals-15-00525-f005:**
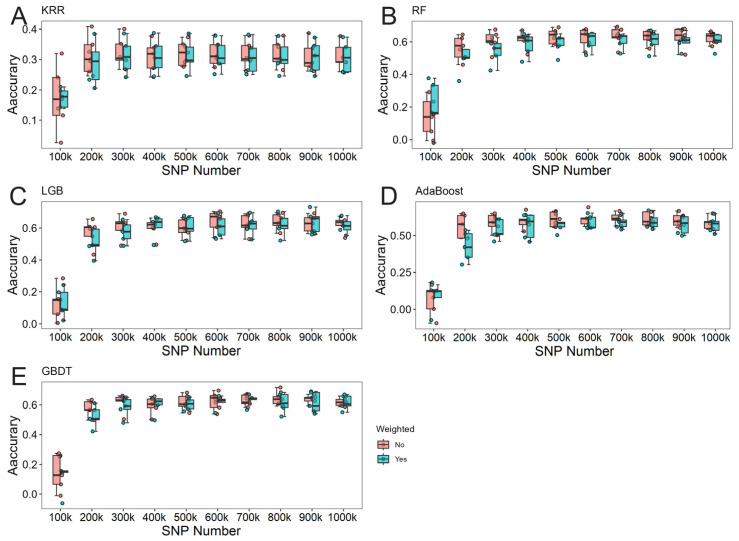
Results comparing weighted and unweighted predictions for total number of piglets born (TNB) in Rongchang pigs. (**A**) The predictive results of KRR. (**B**) The predictive results of RF. (**C**) The predictive results of LightGBM. (**D**) The predictive results of Adaboost. (**E**) The predictive results of GBDT.

**Figure 6 animals-15-00525-f006:**
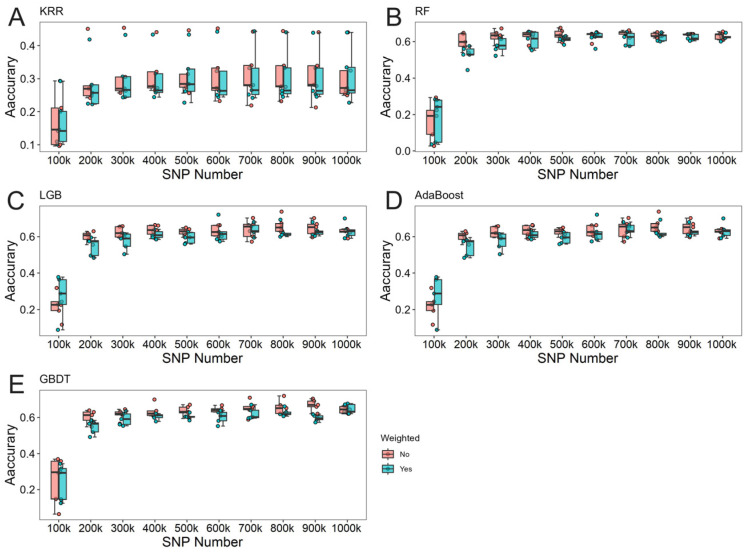
Results comparing weighted and unweighted predictions for number of piglets born alive (NBA) in Rongchang pigs. (**A**) The predictive results of KRR. (**B**) The predictive results of RF. (**C**) The predictive results of LightGBM. (**D**) The predictive results of Adaboost. (**E**) The predictive results of GBDT.

**Figure 7 animals-15-00525-f007:**
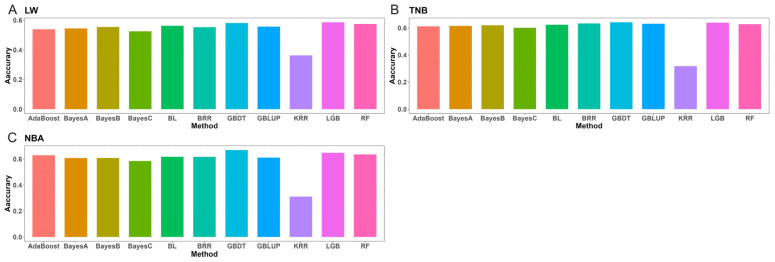
The predictive accuracy of different methods during the independent test for litter weight (LW), total number of piglets born (TNB), and number of piglets born alive (NBA) with the best results in Rongchang pigs. (**A**) The predictive accuracy of LW with different methods. (**B**) The predictive accuracy of TNB with different methods. (**C**) The predictive accuracy of NBA with different methods.

**Figure 8 animals-15-00525-f008:**
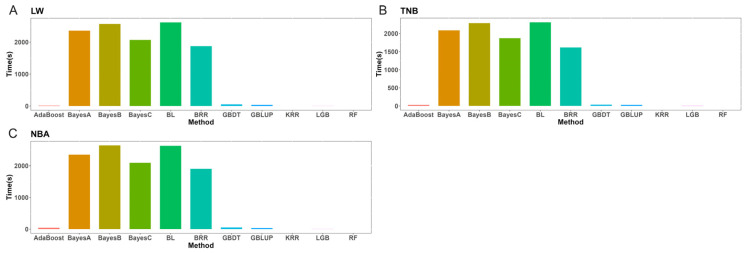
The computational efficiency of different methods for litter weight (LW), total number of piglets born (TNB), and number of piglets born alive (NBA) with the best results in Rongchang pigs. (**A**) The computational efficiency of LW with different methods. (**B**) The computational efficiency of TNB with different methods. (**C**) The computational efficiency of NBA with different methods.

**Table 1 animals-15-00525-t001:** The priori conditions for the Bayes methods.

Method	Priori Conditions
Bayes A	$p\left(\beta_{j},\sigma_{\beta_{j}}^{2},S_{\beta }\right)=\{\prod_{k}N(\beta_{jk}|0,\sigma_{\beta_{jk}}^{2})\chi^{-2}(\sigma_{\beta_{jk}}^{2}|df_{\beta },S_{\beta })\}G(S_{\beta }|r,s)$
Bayes B	$p\left(\beta_{j},\sigma_{\beta_{j}}^{2},\pi \right)=\{\prod_{k}\left[\pi N\left(\beta_{jk}|0,\sigma_{\beta }^{2}\right)+\left(1-\pi \right)1\left(\beta_{jk}=0\right)]\chi^{-2}(\sigma_{\beta_{j}}^{2}|df_{\beta },S_{\beta })\right\}B(\pi |p_{0},\pi_{0})\times G(S_{\beta }|r,s)$
Bayes C	$p\left(\beta_{j},\sigma_{\beta_{j}}^{2},\pi \right)=\{\prod_{k}\left[\pi N\left(\beta_{jk}|0,\sigma_{\beta }^{2}\right)+\left(1-\pi \right)1(\beta_{jk}=0]\}\times \chi^{-2}\left(\sigma_{\beta }^{2}|df_{\beta },S_{\beta }\right)B(\pi |p_{0},\pi_{0}\right)$
BL	$p\left(\beta_{j},\tau_{j}^{2}|\sigma_{\varepsilon }^{2},\lambda \right)=\{\prod_{k}N\left(\beta_{jk}|0,\tau_{jk}^{2}\times \sigma_{\varepsilon }^{2}\right)Exp\{\tau_{jk}^{2}|\frac{\lambda^{2}}{2}\}\}$
BRR	$p\left(\beta_{j},\sigma_{\beta_{j}}^{2}\right)=\{\prod_{k}N(\beta_{jk}|0,\sigma_{\beta }^{2})\}\chi^{-2}(\sigma_{\beta }^{2}|df_{\beta },S_{\beta })$

$N\left(\cdot \right|\cdot ,\cdot )$, $\chi^{-2}\left(\cdot \right|\cdot ,\cdot )$, $G\left(\cdot \right|\cdot ,\cdot )$, $Exp\left(\cdot \right|\cdot ,\cdot )$, and $B\left(\cdot \right|\cdot ,\cdot )$ denote normal, scaled inverse Chi-squared, gamma, exponential, and beta densities, respectively.

**Table 2 animals-15-00525-t002:** Summary of three reproduction traits in Rongchang pigs.

Trait	Number	Mean ± SD	Max	Min	h2(SE)
LW	515	8.11 ± 2.35	15.25	2.55	0.19(0.078)
TNB	515	10 ± 2.51	17	3	0.21(0.080)
NBA	515	10 ± 2.49	17	3	0.15(0.072)

Note: LW, TNB, and NBA represent litter weight, total number of piglets born, and number of piglets born alive, respectively. This notation is consistent throughout the text.

**Table 3 animals-15-00525-t003:** The analysis of variance of three reproduction traits in Rongchang pigs.

Source of Variation	SS	df	MS	F	*p*-Value	F Crit
Between Groups	1356.629	2	678.315	110.857	1.04 × 10^−45^	3.002
Within Groups	9435.241	1542	6.119			
Total	10,791.870	1544				

**Table 4 animals-15-00525-t004:** The candidate genes of three reproduction traits in Rongchang pigs.

Trait	Chr	Candidate Gene	Starting Physical Position/bp	Terminate Physical Position/bp
LW	12	*GAS7*	54,752,507	54,973,515
	11	*HS6ST3*	65,516,097	66,187,242
	12	*GSG1L2*	54,665,268	54,677,502
	15	*EFHD1*	133,120,093	133,169,428
	6	*WWOX*	8,910,138	9,871,053
	13	*STXBP5L*	139,123,209	139,445,470
	13	*U2*	139,311,331	139,311,450
	12	*RARA*	22,047,443	22,085,673
	15	*DNER*	130,415,061	130,754,048
	3	*SDK1*	2,814,547	3,324,798
	13	*APP*	189,435,094	189,716,056
	7	*SLC7A7*	76,198,256	76,235,402
	6	*MEIOSIN*	52,079,224	52,095,099
	6	*DMPK*	52,100,259	52,110,478
	1	*ENTREP1*	222,658,382	222,725,259
	13	*CYYR1*	189,957,611	190,066,539
	14	*PALLD*	20,678,452	21,020,125
	9	*NMNAT2*	124,498,109	124,696,194
	14	*PTPRE*	137,101,555	137,264,238
	8	*KCNIP4*	15,211,917	16,346,786
	3	*ALMS1*	69,253,750	69,417,189
	17	*ARFGEF2*	50,631,092	50,732,243
	2	*SPOCK1*	139,018,179	139,522,593
	9	*ME3*	20,294,908	20,583,584
	1	*MAMDC2*	223,384,827	223,548,420
	16	*ZNF622*	5,699,744	5,714,516
	13	*POLQ*	138,978,609	139,108,500
	3	*LRATD1*	122,716,040	122,731,791
	7	*SLC7A8*	75,854,882	75,921,931
	12	*TTLL6*	24,987,347	25,022,530
	1	*NCOA7*	37,043,805	37,193,958
	18	*LEP*	20,106,868	20,123,323
TNB	14	*PALLD*	20,678,452	21,020,125
	15	*EFHD1*	133,120,093	133,169,428
	1	*PRKN*	5,465,312	6,730,872
	12	*ARHGAP44*	57,049,895	57,206,889
	9	*NARS2*	12,867,686	13,010,887
	8	*LDB2*	11,641,064	12,037,265
	9	*GAB2*	12,663,657	12,851,342
	14	*PRF1*	73,512,769	73,520,202
	1	*LAMC3*	271,004,433	271,071,223
	5	*LMNTD1*	47,848,084	48,312,305
	14	*LRMDA*	77,957,851	79,051,455
	9	*HHAT*	132,347,037	132,666,873
	8	*KCNIP4*	15,211,917	16,346,786
	15	*DIS3L2*	132,540,829	132,906,232
	9	*CD34*	134,913,121	134,937,762
	14	*PALD1*	73,401,018	73,483,354
NBA	14	*PALLD*	20,678,452	21,020,125
	15	*EFHD1*	133,120,093	133,169,428
	12	*TTLL6*	24,987,347	25,022,530
	12	*ARHGAP44*	57,049,895	57,206,889
	14	*CFAP58*	115,285,979	115,410,481
	9	*NARS2*	12,867,686	13,010,887
	6	*DPY19L3*	42,334,736	42,414,830
	5	*LMNTD1*	47,848,084	48,312,305
	13	*GRIK1*	192,877,606	193,294,952

**Table 5 animals-15-00525-t005:** The correlation coefficients between cross-validation and independent tests of different methods in Rongchang pigs.

Method	Traits		
	LW	TNB	NBA
KRR	0.998	0.971	0.983
RF	0.995	0.995	0.993
LGB	0.935	0.990	0.978
AdaBoost	0.976	0.993	0.985
GBDT	0.993	0.997	0.971
Bayes A	0.991	0.997	0.993
Bayes B	0.995	0.993	0.998
Bayes C	0.977	0.997	0.992
BL	0.996	0.997	0.998
BRR	0.989	0.994	0.998
GBLUP	0.995	0.996	0.995

Note: KRR, RF, LGB, GBDT, BL, and BRR represent kernel ridge regression, random forest, Gradient Boosting Decision Tree, Light Gradient Boosting Machine, Bayes LASSO, and Bayes Ridge Regression, respectively.

## Data Availability

The original contributions presented in the study are included in the article, further inquiries can be directed to the corresponding author.

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
