# Peer review of "Enhancing Genomic Prediction Accuracy of Reproduction Traits in Rongchang Pigs Through Machine Learning"

_animals, 2025, doi:10.3390/ani15040525_

Round 1

Reviewer 1 Report

Comments and Suggestions for Authors

The manuscript needs revision. Please refer to comments given in the text of reviewed attached file of the manuscript.

Author Response

Dear Reviewer:

Thanks very much for taking your time to review this manuscript. I really appreciate all your comments and suggestions! Please find my itemized responses in below and my revisions/corrections in the re-submitted files.

Comments 1: What is the basic problem that your research focuses on and is done to solve?

Response 1: We focus on using machine learning methods to improve the accuracy of genomic prediction for the local pigs.

Comments 2: Please specify the main knowledge gap that your article has filled in the text.

Response 2: We use imputation from 50K chip to whole-genome sequencing data and find the appropriate ML models and parameters for Rongchang pigs.

Comments 3: This conclusion is wide and general, please add your specific conclusion from your specific results at the end of abstract.

Response 3: We rewritten abstract and emphasized the effect of GBDT model. (At line 39-line 41).

Comments 4: It is better to explain about application an important of using AI in genomics and animal breeding. For this, you can use below sentences and references:

Response 4: We agree with this comment and have added the sentences and references. (At line 91-line 106)

Comments 5: Please specify in the objective whether your research is being conducted for the first time in the world or is it a continuation of another research?

Response 5: Our research is the first time in the world for Rongchang pigs.

Comments 6: What is the superiority of your research compared to other researches?

Response 6: We used whole-genome sequencing data and weighted design based on GWAS results.

Comments 7: Please add country.

Response 7: It has been added at line 125.

Comments 8: Please add company name and country for used devices and kits.

Response 8: It has been added at line 143 and line 149.

Comments 9: Please identify sex of used animals in the text of the manuscript.

Response 9: It has been added at line 139. All animals of the manuscript is female.

Comments 10: It is better to compare your results with the results of other researchers who used ML in animal genome studies and discuss more.

Response 10: We added a of the comparison with similar study at line 557-line 562, and emphasized the effect of GWAS results.

The statements have been corrected. We will be happy to edit the text further based on helpful comments from you, and thanks again for your comments to this manuscript.

Reviewer 2 Report

Comments and Suggestions for Authors

Dear Authors,

I have carefully reviewed your manuscript titled "Enhancing Genomic Prediction Accuracy of Reproduction Traits in Rongchang Pigs through Machine Learning." Your study presents a timely and relevant approach to genomic selection, comparing traditional genomic prediction methods with machine learning (ML) models. This research is particularly important for advancing livestock breeding programs through computational approaches.

While your manuscript is scientifically relevant and methodologically sound, there are several areas that require significant revision to improve clarity, depth, and impact. Below, I provide detailed feedback and recommendations for improving your manuscript.

Strengths of the Manuscript

- Your study applies ML models to genomic prediction, a cutting-edge approach in livestock genetics.

- The manuscript compares multiple traditional and ML models, strengthening the validity of your findings.

- The inclusion of cross-validation and independent testing enhances the reliability of your results.

Major Areas for Improvement

1. Lack of Biological Interpretation of GWAS Results

- Your manuscript reports significant SNP loci for reproduction traits but does not discuss their biological relevance.

Recommended:

Perform gene annotation to identify potential candidate genes near significant SNPs.

Discuss whether these genes are known to influence reproduction traits in pigs.

Compare your findings with previous GWAS studies on pig reproduction to evaluate consistency and novelty.

2. Insufficient Justification for Machine Learning Model Selection

- You apply five ML models but do not explain why these specific models were chosen.

Recommended:

Justify why tree-based models (RF, GBDT) were prioritized over other ML models.

Explain why Support Vector Machines (SVM) or Neural Networks were not included.

Discuss how model complexity and feature selection affect prediction accuracy.

3. No Discussion on Overfitting and Model Robustness

- You mention that ML models overfit NBA when SNPs exceed 600k, but no solutions are provided.

Recommended:

Discuss how hyperparameter tuning, LASSO regularization, or feature selection could reduce overfitting.

Provide a feature importance ranking for ML models to show which SNPs contributed most to predictions.

4. Computational Efficiency Trade-offs Are Not Analyzed

- You claim that ML models are computationally faster than Bayesian methods, but you do not analyze whether this affects prediction accuracy.

Recommended:

Provide a trade-off analysis between computational efficiency and prediction accuracy.

Discuss whether ML models remain scalable for larger datasets.

5. Conclusion Lacks Depth and Future Directions

- Your conclusion is too general and does not discuss study limitations or future research directions.

 Recommended:

Expand the conclusion to discuss how these findings apply to pig breeding programs.

Acknowledge study limitations (e.g., sample size, SNP weighting biases, potential overfitting issues).

Summary of Recommended Revisions

Key Revisions Needed:

Provide gene annotation and biological interpretation of significant SNPs.

Justify the selection of ML models and compare their strengths/weaknesses.

Perform statistical validation for phenotypic data using ANOVA or t-tests.

Address overfitting concerns and discuss feature importance in ML models.

Analyze computational efficiency trade-offs between ML and traditional methods.

Expand the conclusion with study limitations and future research directions.

Final Recommendation: Major Revisions Before Reconsideration

While your manuscript presents valuable findings, it requires major revisions before it can be reconsidered for acceptance. The biological interpretation, methodological justification, statistical validation, and computational analysis must be strengthened to improve clarity and scientific impact.

Once these revisions are made, your manuscript will be substantially improved and more suitable for publication. I appreciate your efforts and look forward to reviewing the revised manuscript.

Sincerely,

Author Response

Dear Reviewer:

Thanks very much for taking your time to review this manuscript. I really appreciate all your comments and suggestions! Please find my itemized responses in below and my revisions/corrections in the re-submitted files.

Comments 1: Provide gene annotation and biological interpretation of significant SNPs.

Response 1: We added the gene annotation of significant SNPs at line 352-line 359, and added the discussion at line 469-line 480.

Comments 2: Justify the selection of ML models and compare their strengths/weaknesses.

Response 2: We added the discussion at line 481-line 491, focused on the advantages of tree-based machine learning and the impact of sample size on neural networks.

Comments 3: Perform statistical validation for phenotypic data using ANOVA or t-tests.

Response 3: We provide ANOVA of phenotypic data and discussion at line 317, line 461.

Comments 4: Address overfitting concerns and discuss feature importance in ML models.

Response 4: We added the discussion at line 571-line 575, include hyperparameters, sample size and weights.

Comments 5: Analyze computational efficiency trade-offs between ML and traditional methods.

Response 5: We added the discussion at line 593-line 596, to illustrate the advantages of machine learning in high SNP density.

Comments 6: Expand the conclusion with study limitations and future research directions.

Response 6: We added study limitations and future research directions at line 605-line 607.

The statements have been corrected. We will be happy to edit the text further based on helpful comments from you, and thanks again for your comments to this manuscript.

Reviewer 3 Report

Comments and Suggestions for Authors

Manuscript „Enhancing Genomic Prediction Accuracy of Reproduction Trait in Rongchang Pigs through Machine Learning“ is interesting but I have few comments and questions to make improved and make suitable for publication in Animals journal. The topic is relevant and interesting considering the growing interest for machine learning in animal science.

Abstract

This section is too long. It must be corrected and adjusted to the Instruction for authors (no more the 200 words). Also, there are'n any key words. It should be added. Moreover, Authors should write Simple summary.

Introduction

References are not cite in appropriate way in whole manuscript. Please check the Instruction for authors and correct them.

Introduction is quite long and concise, but there should be briefly described and compared application of machine learning in genomic prediction for reproduction traits for livestock emphazing similarities and gaps. Please use other recently published paper for enhancing this part of the manuscript.

L98-L99 – This must be enhanced, highlight and clear the novelty of Your study, and its importance.

Material and methods

L100 – correct the section title

Subsection title is “Animals and phenotype“ but information about phenotype is not described. Provide informations on diet, housing and rearing. Specify the software for correction model. Add additional information on quality assurance methods. Please, better explanation on GWAS and QTL integration.

Results / discussion

L295 – correct the Table 2, provide additional information of used abervation for breeds. Also, describe results of table, in section „Discussion“ wright a comparison with other studies.

Authors identified significant loci on SSC12, SSC14, and SSC15. I suggest to discuss and described the biological importance of this. Also, generate Manhattan plot to visualise the GWAS results.

In section „Accuracy of genomic prediction in CV„  discuss why these models outperform traditional methods at higher SNP densities. Also, i suggest to correct the section title and not use the abervations in it. Correct that in whole manuscript.

L364 – delete the space before the subtitle

L379 – correct the table as comment for Table 2.

L376-L378 – move and place it between the Table 2 and Figure 7.

L398 – Figure is unclear and obtained results is not shown in clear way

Discussion is well written, however there are some thing that sholud be corrected and more concise described. There should be santance on the limitation of analysed method (machine learning), and the limitations of the weighting approach for complex traits.

Also, I suggest to wright final sentance on application of these models in real sector, in breeding programes as well as future work and possible limitation.

Conclusion

Conclusion is well written.

References

References is not written in accordance to Template and Instruction for authors and must be corrected.

Author Response

Dear Reviewer:

Thanks very much for taking your time to review this manuscript. I really appreciate all your comments and suggestions! Please find my itemized responses in below and my revisions/corrections in the re-submitted files.

Comments 1: Abstract: This section is too long. It must be corrected and adjusted to the Instruction for authors (no more the 200 words). Also, there are'n any key words. It should be added. Moreover, Authors should write Simple summary.

Response 1: We have rewritten the abstract to ensure it meets the requirements, and added the Simple summary and key words at line 12, line 43, respectively.

Comments 2: Introduction: References are not cite in appropriate way in whole manuscript. Please check the Instruction for authors and correct them.

Response 2: We have re-edited the manuscript according to the manuscript template file to address the issues.

Comments 3: Introduction is quite long and concise, but there should be briefly described and compared application of machine learning in genomic prediction for reproduction traits for livestock emphazing similarities and gaps. Please use other recently published paper for enhancing this part of the manuscript.

Response 3: We enhancing this part at line 80-line 84.

Comments 4: L98-L99 – This must be enhanced, highlight and clear the novelty of Your study, and its importance.

Response 4: We emphasized the challenges of small population size and SNP complexity at line 118-line 121.

Comments 5: L100 – correct the section title.Subsection title is “Animals and phenotype“ but information about phenotype is not described. Provide informations on diet, housing and rearing. Specify the software for correction model. Add additional information on quality assurance methods. Please, better explanation on GWAS and QTL integration.

Response 5: We have sspecify the software (R 4.3.1) for correction model at line 130, and provide phenotype informations at line 311-line 323.

Comments 6: Results / discussion: L295 – correct the Table 2, provide additional information of used abervation for breeds. Also, describe results of table, in section „Discussion“ wright a comparison with other studies.

Response 6: We provide additional information of used abervation at line 315 ,and added the discussion at line 557-line 562.

Comments 7: Authors identified significant loci on SSC12, SSC14, and SSC15. I suggest to discuss and described the biological importance of this. Also, generate Manhattan plot to visualise the GWAS results.

Response 7: We added the gene annotation of significant SNPs at line 352-line 359, and added the discussion at line 469-line 480.

Comments 8: In section „Accuracy of genomic prediction in CV„  discuss why these models outperform traditional methods at higher SNP densities. Also, i suggest to correct the section title and not use the abervations in it. Correct that in whole manuscript.

Response 8: Abervations of CV has been corrected that in whole manuscript.

Comments 9: L364 – delete the space before the subtitle

Response 9: It has been deleted.

Comments 10: L379 – correct the table as comment for Table 2.

Response 10: We provide additional information of used abervation at line 417.

Comments 11: L376-L378 – move and place it between the Table 2 and Figure 7.

Response 11: It has been moved.

Comments 12: Figure is unclear and obtained results is not shown in clear way.

Response 12: We enhanced the dpi of Figure 8 at line 437.

Comments 13: References: References is not written in accordance to Template and Instruction for authors and must be corrected.

Response 13: We have re-edited the manuscript according to the manuscript template file to address the issues.

The statements have been corrected. We will be happy to edit the text further based on helpful comments from you, and thanks again for your comments to this manuscript.

Round 2

Reviewer 2 Report

Comments and Suggestions for Authors

Dear Authors,

 I have carefully reviewed the revised manuscript and am pleased to note that the requested improvements have been successfully implemented. I have no further comments or suggestions, and I fully support accepting the manuscript for publication in its current form.

 Best regards,